# CatVTON: Concatenation Is All You Need for Virtual Try-On with Diffusion Models

**Zheng Chong[1,4,5], Xiao Dong[2], Haoxiang Li[3], Shiyue Zhang[1], Wenqing Zhang[1], Xujie Zhang[1], Hanqing Zhao[4,6], Dongmei Jiang[4] & Xiaodan Liang[1,4,7]***

[1]Shenzhen Campus of Sun Yat-sen University, Shenzhen, Guangdong 518107, P.R. China
[2]School of Artificial Intelligence, Sun Yat-sen University, Zhuhai Campus, Zhuhai 519082, China
[3]Pixocial Labs     [4]Pengcheng Laboratory     [5]LavieAI
[6]Shenzhen Institute of Advanced Technology, Chinese Academy of Sciences, Shenzhen, China
[7]Guangdong Key Laboratory of Big Data Analysis and Processing, Guangzhou, 510006, China
`{chongzheng98,dx.icandoit}@gmail.com`, `haoxiang@pixocial.com`,
`{zhangshy,zhangwq76,zhangxj59}@mail2.sysu.edu.cn`,
`hq.zhao79@gmail.com`, `jiangdm@pcl.ac.cn`, `xdliang328@gmail.com`

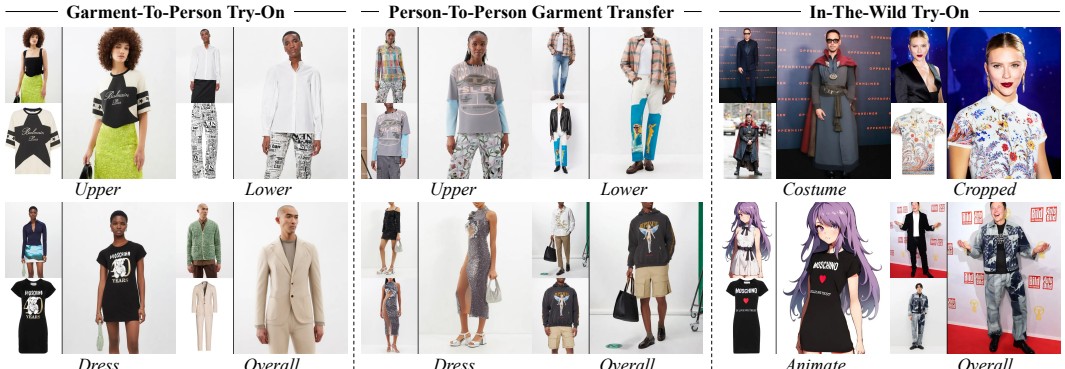

Figure 1: CatVTON enables transferring in-shop or worn garments to the target person across various categories. With a lightweight architecture and efficient training (49.57M parameters, trained on 73K samples), our model allows inference without additional preprocessing, delivering high-quality virtual try-ons with fine-grained consistency in challenging scenarios like comics, complex backgrounds, special garments, and cropped images.

## Abstract

Virtual try-on methods based on diffusion models achieve realistic effects but often require additional encoding modules, a large number of training parameters, and complex preprocessing, which increases the burden on training and inference. In this work, we re-evaluate the necessity of additional modules and analyze how to improve training efficiency and reduce redundant steps in the inference process. Based on these insights, we propose CatVTON, a simple and efficient virtual try-on diffusion model that transfers in-shop or worn garments of arbitrary categories to target individuals by concatenating them along spatial dimensions as inputs of the diffusion model. The efficiency of CatVTON is reflected in three aspects: (1) Lightweight network. CatVTON consists only of a VAE and a simplified denoising UNet, removing redundant image and text encoders as well as cross-attentions, and includes just 899.06M parameters. (2) Parameter-efficient training. Through experimental analysis, we identify self-attention modules as crucial for adapting pre-trained diffusion models to the virtual try-on task, enabling high-quality results with only 49.57M training parameters. (3) Simplified inference. CatVTON eliminates unnecessary preprocessing, such as pose estimation, human parsing, and captioning, requiring only person image and garment reference to guide the virtual try-on process, reducing 49%+ memory usage com-

pared to other diffusion-based methods. Extensive experiments demonstrate that CatVTON achieves superior qualitative and quantitative results compared to baseline methods and demonstrates strong generalization performance in in-the-wild scenarios, despite being trained solely on public datasets with 73K samples.

# 1 INTRODUCTION

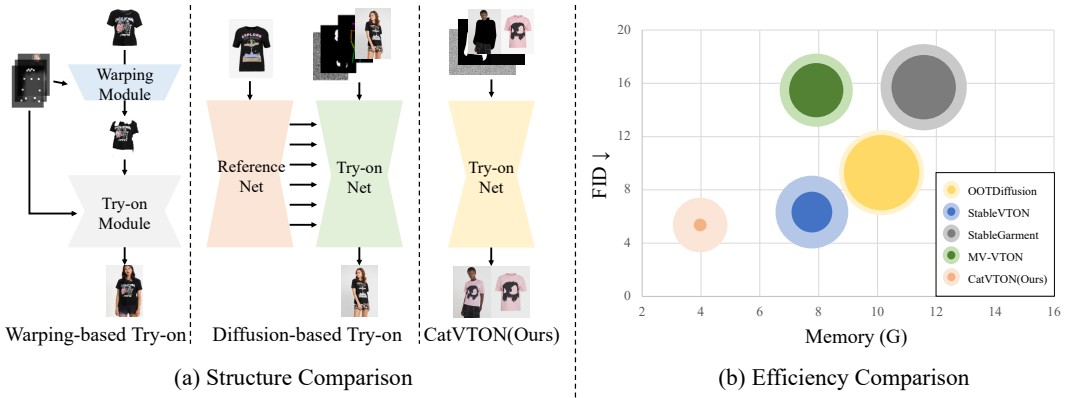

(a) Structure Comparison          (b) Efficiency Comparison

Figure 2: (a) Structure comparison of different try-on methods. CatVTON eliminates the need for garment warping or additional ReferenceNet resulting in a simple structure. (b) Efficiency comparison with diffusion-based try-on methods. Each method is represented by two concentric circles, where the outer circle denotes the total parameters and the inner circle indicates the trainable parameters. CatVTON achieves lower FID on the VITON-HD dataset with fewer total parameters, trainable parameters, and memory usage.

Virtual Try-On (VTON), which transfers specific garments onto user photos, has attracted considerable interest due to its potential applications in e-commerce. Early try-on methods (Han et al., 2018; Wang et al., 2018; Han et al., 2019; Minar et al., 2020; Ge et al., 2021; Xie et al., 2021b) employ a two-stage process of pose-guided garment warping followed by blending with the target person, as illustrated in the left of Figure 2 (a). However, these methods often result in unnatural fits and struggle with complex poses due to the limited warping process.

Benefitting from the success of diffusion models (Rombach et al., 2021), many diffusion-based try-on methods (Zhu et al., 2023; Kim et al., 2023; Xu et al., 2024; Morelli et al., 2023; Choi et al., 2024; Wang et al., 2024c; xujie zhang et al., 2023; Sun et al., 2024) have emerged and achieved more natural try-on results. As shown in the middle of Figure 2 (a), these methods adopt a structure called Dual-UNet or ReferenceNet for processing garment images. Some methods (Kim et al., 2023; Choi et al., 2024; Xu et al., 2024; Sun et al., 2024) also integrate image encoders, such as CLIP (Radford et al., 2021) and DINOv2 (Oquab et al., 2023), to capture additional garment features. However, these encoders contribute to a more complex and computationally intensive network architecture, increasing the burdens of both training and inference.

To integrate diffusion models into virtual try-on systems without sacrificing efficiency, it is essential to discuss the role of extra image encoders and ReferenceNet. Pre-trained image encoders like DINOv2 and CLIP are not optimized for detail preservation—a crucial factor in virtual try-on applications. In contrast, ReferenceNet, by replicating the structure and weights of the backbone UNet, allows for the generation of multi-scale garment features that naturally share latent spaces with the backbone layers. This feature-sharing facilitates a seamless link between garment and person representations, improving the overall accuracy of the virtual try-on process. Based on this shared latent space mechanism, we realized that the model architecture could be further simplified. If the garment and person features can be efficiently integrated within the shared latent space, is it possible to use a single UNet model to process both person and garment images simultaneously? Such an approach would not only eliminate redundant encoders but also enhance try-on system efficiency by streamlining the model.

Building on this, we propose CatVTON, a simple and efficient diffusion-based virtual try-on model. Our CatVTON removes unnecessary encoders, and streamlines the garment and person interaction,

thereby enabling efficient training and inference. Specifically, as shown in Figure 3, our model comprises only a VAE for mapping images to the latent space and a simplified UNet for denoising from LDM (Rombach et al., 2021). We further remove the text encoder and the cross-attention modules as text conditions are not essential for try-on, simplifying the architecture to a total of 899.06M parameters. To optimize training efficiency, we investigated the effective modules in UNet to interact with garment and person features. By progressively adjusting the trainable modules in experiments, we find that self-attention modules with a global receptive field (Dosovitskiy et al., 2021) are the most critical part for try-on task with diffusion models, and achieve realistic try-on results by training only 49.57M parameters. Furthermore, we explored a more straightforward and efficient inference process. Numerous try-on methods (Sun et al., 2024; Zhang et al., 2024b; Wang et al., 2024c; Choi et al., 2024; Kim et al., 2023) depend on extra preprocessing such as pose estimation, human parsing, and captioning to guide the try-on process, thereby increasing the computational burden during inference. Hence, we think that the garment and person images contain sufficient information for try-ons, and removing additional conditions can simplify the model while achieving efficient try-ons without compromising quality. By integrating these enhancements, CatVTON outperforms other diffusion-based try-on methods in both effectiveness and efficiency, as shown in Figure 2 (b) and Table 3.

In summary, the contributions of this work include:

- We propose CatVTON, a lightweight virtual try-on diffusion model with only 899.06M parameters, that achieves high-quality results by simply concatenating garment and person images as inputs, eliminating the need for extra image encoders, ReferenceNet, and text-conditioned modules.
- We introduce a parameter-efficient training strategy to transfer pre-trained diffusion models to virtual try-on tasks while preserving prior knowledge by training necessary modules with only 49.57M parameters.
- We simplify the inference process by eliminating the need for extra pre-processing of input images and leverage the robust priors from pre-trained diffusion models to infer all necessary information, reducing memory usage by 49%+ compared to other diffusion-based baselines.
- Extensive experiments on the VITON-HD and DressCode datasets demonstrate that our method produces high-quality virtual try-on results with consistent details, outperforming state-of-the-art baselines in qualitative and quantitative analyses, and performs well in in-the-wild scenarios.

## 2 RELATED WORK

### 2.1 SUBJECT-DRIVEN IMAGE GENERATION

Subject-driven image generation is a hot topic in the field of image generation, focusing on integrating the target subject into new scenes or perspectives while maintaining consistency with the subject. LoRA (Hu et al., 2021) and DreamBooth (Ruiz et al., 2022) train individual models for each subject, achieving consistent subject-driven generation, but the frequent training incurs a high cost. Paint by Example (Yang et al., 2022) and IP-Adapter (Ye et al., 2023) leverage CLIP (Radford et al., 2021) image encoders to extract subject features and inject them into diffusion models via cross-attention, enabling convenient subject-driven generation. However, they fall short of preserving details. In contrast, AnyDoor (Chen et al., 2023) employs DINOv2 (Oquab et al., 2023) and ControlNet (Zhang et al., 2023) to jointly extract subject features to achieve more accurate subject-driven image generation. PCDMs (Shen et al., 2024) achieve high consistency in transferring persons to different perspectives through three progressive diffusion models. InstantID (Wang et al., 2024b) introduces an additional IdentifyNet to encode facial information, achieving high-fidelity facial stylization. Similarly, MimicBrush (Ju et al., 2024) proposes a dual-branch model that learns from video data and masked image modeling to accomplish subject-driven generation. While these methods achieve high-quality subject-driven generation, they also lead to complex network architectures and a large number of trainable parameters, which limit their applications.

### 2.2 IMAGE-BASED VIRTUAL TRY-ON

In image-based virtual try-on, the goal is to create a composite image of a person wearing a specified garment while maintaining identity and consistency. Warping-based methods typically decompose the task into two stages: garment warping and fusion based on warped garments. Some

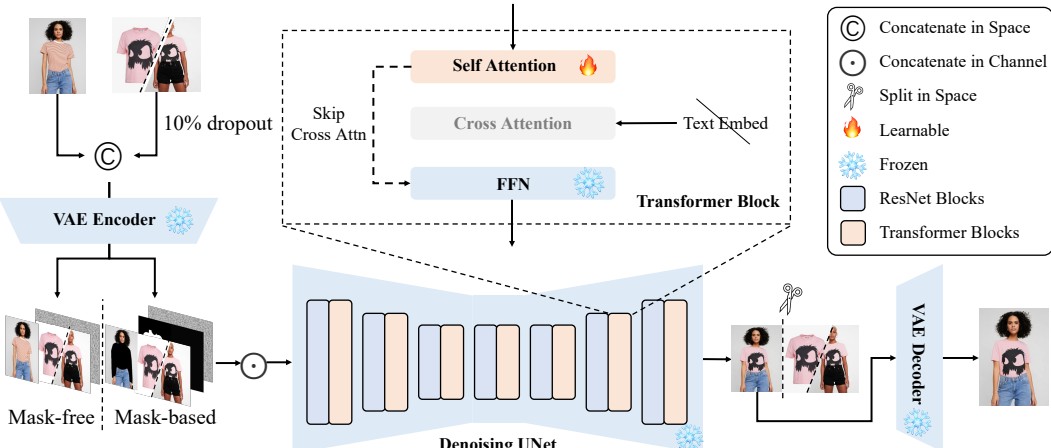

Figure 3: Overview of CatVTON. Our method achieves high-quality try-ons by simply concatenating the conditional image (garment or reference person) with the target person image in the spatial dimension, ensuring they remain in the same feature space during denoising. Only self-attention parameters, which provide global interaction, are learnable, while cross-attention for text interaction is omitted. No additional conditions (pose, parsing) are needed, resulting in a lightweight network with minimal trainable parameters and simplified inference.

warping-based methods (Wang et al., 2018; Han et al., 2018; Choi et al., 2021) utilize geometric deformation like TPS (Bookstein, 1989) to warp the garment, while others (Han et al., 2019; Ge et al., 2021; Xie et al., 2021b; 2023; Gou et al., 2023) estimate an appearance flow map to model non-rigid deformation for more complex garment warping. Besides, PASTA-GANs (Xie et al., 2022; 2021a) propose a patch-routed disentanglement module for pose-guided garment warping. However, warping-based methods often struggle with alignment issues caused by inaccurate TPS or flow estimation. Diffusion-based methods leverage the generation capacity of pre-trained diffusion models to avoid the limitations of garment warping. LaDI-VTON (Morelli et al., 2023) and StableVITON (Kim et al., 2023) employ a ControlNet-like structure to encode additional information. TryOnDiffusion (Zhu et al., 2023) designs two UNets for feature extraction of garment and person images, respectively, and achieves impressive results. BoowVTON (Zhang et al., 2024a) utilizes generated pseudo data to train the diffusion model and employs a clothing encoder to provide garment information, achieving mask-free virtual try-on. OOTDiffusion (Xu et al., 2024), StableGarment (Wang et al., 2024c), IDM-VTON (Choi et al., 2024), and OutfitAnyone (Sun et al., 2024) utilize a ReferenceNet structure, similar to the denoising UNet from pre-trained models, to process garment images, with slight structural variations. However, these methods often require complex network structures, numerous trainable parameters, and various conditions to assist inference, which inspires our exploration towards efficient virtual try-on diffusion models.

## 3 METHODS

CatVTON aims to streamline diffusion-based virtual try-on methods by eliminating redundant components, focusing on key modules, and simplifying preprocessing requirements.

### 3.1 LIGHTWEIGHT NETWORK

Our lightweight structure arises from the consideration of image representations for garments and persons and their effective interaction. Recent studies (Ye et al., 2023; Chen et al., 2023) have demonstrated that existing pre-trained encoders, such as DINOv2 (Oquab et al., 2023) and CLIP (Radford et al., 2021), struggle to preserve fine details for subject-driven image generation. This indicates that using these encoders to encode garment images for try-on purposes is insufficient, hence we remove all additional image encoders in our method. Methods with ReferenceNet enhance detailed alignment in diffusion-based try-on by replicating weights from a denoising UNet and performing fine-tuning. However, this approach introduces additional trainable modules and increases the computational load. To address this, we concatenate person and garment images along the spatial dimension as inputs to the original denoising UNet to avoid importing any new modules.

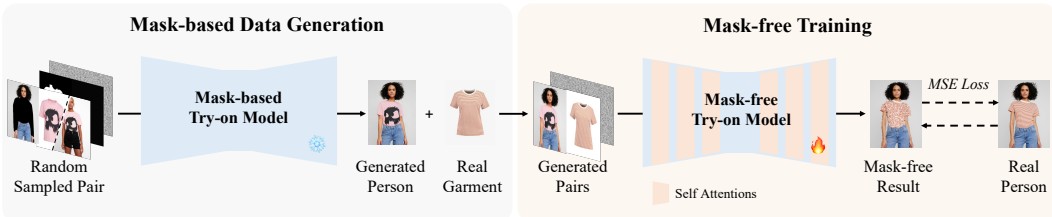

Figure 4: Overview of the mask-free training pipeline. We first use the trained mask-based model to generate synthetic person image from randomly sampled person-garment pairs. These synthetic person images, along with their corresponding original person and garment images, form the training data for the mask-free model.

As shown in Figure 3, CatVTON features a lightweight network structure comprising only two essential modules: **(1) VAE**. The VAE encoder encodes the person and garment images into the latent space, optimizing computational efficiency during diffusion. Once encoded, the latent garment and person are concatenated in the spatial dimension as inputs to the denoising UNet. Then, the VAE decoder reconstructs the output latent into the original pixel space after denoising. **(2) Simplified Denoising UNet**. As the text condition is not necessary for image-based try-on tasks and our experiment reveals that training with text conditions leads to a detrimental impact on try-on performance (demonstrated in experiments section), we remove the text encoder and cross-attention modules in the UNet to further simplify the network and reduce 167.02M parameters. The simplified denoising UNet accepts concatenated garments and persons as conditions, along with noise and masks, and generates the predicted try-on latent. Integrating these two modules, the proposed lightweight try-on diffusion model has only 899.06M parameters, representing a reduction of over 44% compared to other diffusion-based methods.

## 3.2 Parameter-Efficient Training

CatVTON aims to optimize the interaction between garment and person features with the fewest trainable modules in LDMs (Rombach et al., 2021) for parameter-efficient training. Diffusion-based methods typically train the entire U-Net to adapt pre-trained models to the virtual try-on task. However, since LDMs have undergone extensive pre-training on large-scale datasets, they already possess robust prior knowledge. When transferring LDMs to the try-on task, it is only necessary to fine-tune the parameters related to the interaction between person and garment features.

As shown in Figure 3, the denoising UNet comprises alternating ResNet (He et al., 2015) and transformer (Vaswani et al., 2023) blocks. The transformer blocks, equipped with self-attention layers for global interaction, complement the ResNet's local feature capture, which stems from its convolutional architecture. We conduct experiments to gradually find the most relevant modules. We set the trainable components to 1) the entire U-Net, 2) the transformer blocks, and 3) the self-attention layers. The results indicate that despite a significant disparity in the number of trainable parameters (815.45M, 267.24M, and 49.57M, respectively), all three variants produced satisfactory virtual try-on results, and no substantial differences are observed in visual quality and metrics among them (detailed in experiments section).

Consequently, we adopted a parameter-efficient training strategy by finetuning only the self-attention layers with 49.57M parameters. For the training of the mask-free try-on model, we first leverage the already trained mask-based model to infer generated person images from randomly sampled person-garment pairs in the same datasets. These generated person images, along with their corresponding original person and garment images, form the training data for the mask-free model, as shown in Figure 4. For both the mask-based and mask-free try-on models, we employ Mean Squared Error (MSE) loss for training. Additionally, we adopt a 10% conditional dropout to support classifier-free guidance (CFG) (Ho & Salimans, 2022) and employ the DREAM (Zhou et al., 2024) strategy during training. The ablation studies of CFG and DREAM are illustrated in the experiments section.

## 3.3 Simplified Inference

Besides training, we also explored a more straightforward and more efficient inference process for image-based try-on. We simplified the inference by eliminating the need for any preprocessing or conditional information. The whole process can be completed with only the person image and

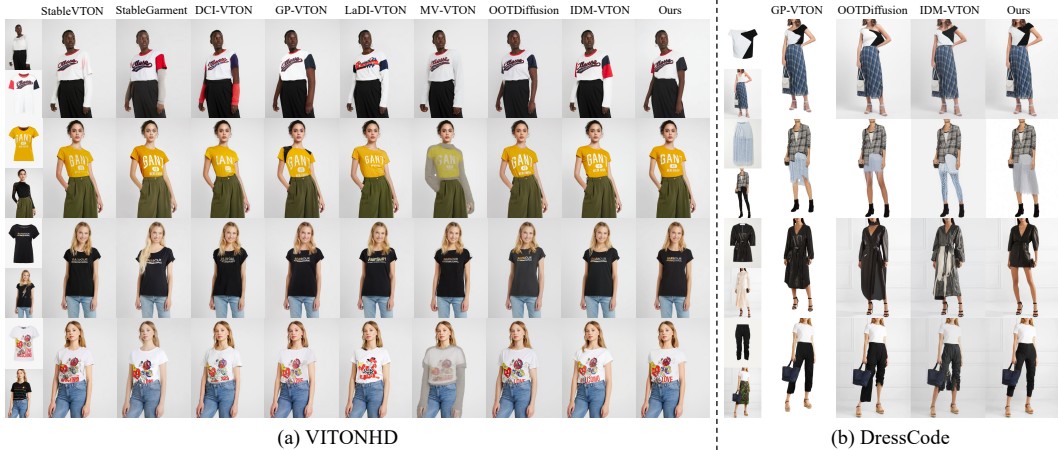

(a) VITONHD         (b) DressCode

Figure 5: Qualitative comparison on the VITON-HD and DressCode dataset. CatVTON demonstrates a distinct advantage in handling complex patterns and text. Please zoom in for more details.

garment reference for the mask-free model and an additional binary mask for the mask-based model. Specifically, given a target person image $I_p \in \mathbb{R}^{3 \times H \times W}$ and a binary cloth-agnostic mask $M \in \mathbb{R}^{H \times W}$, an input person image $I_i$ is obtained by:

$$I_i = \begin{cases} I_p & \text{if mask-free} \\ I_p \otimes M & \text{else} \end{cases}, \tag{1}$$

where $\otimes$ represents the element-wise (Hadamard) product. Then input person image $I_i \in \mathbb{R}^{3 \times H \times W}$ and the garment reference (either in-shop garment or worn person image) $I_g \in \mathbb{R}^{3 \times H \times W}$ is encoded into the latent space by the VAE encoder $\varepsilon$:

$$X_i = \varepsilon(I_i \copyright I_g), \tag{2}$$

where $\copyright$ denotes the concatenation operation along the spatial dimension and $X_i \in \mathbb{R}^{4 \times \frac{H}{8} \times \frac{W}{4}}$. For mask-based model, $M$ is also concatenated with all-zero masks and then interpolated to match the size of latent space, resulting in $m_i \in \mathbb{R}^{\frac{H}{8} \times \frac{W}{4}}$:

$$M_i = Interpolate(M \copyright O), \tag{3}$$

where $O$ represents the all-zero mask with the same size as M. At the beginning of the denoising, the input conditions and a random noise $z_T \sim \mathcal{N}(0,1) \in \mathbb{R}^{4 \times \frac{H}{8} \times \frac{W}{4}}$ of the same size as $X_i$ are concatenated along the channel dimension and input to the denoising UNet to get predicted $z_{T-1}$, and this process is repeated for $T$ times to predict the final latent $z_0$. For denoising step $t$, this process can be written as:

$$z_{t-1} = \begin{cases} \text{UNet}(z_t \odot X_i) & \text{if mask-free} \\ \text{UNet}(z_t \odot M_i \odot X_i) & \text{else} \end{cases}, \tag{4}$$

where $\odot$ denotes the concatenation operation along the channel dimension, finally, $z_0 \in \mathbb{R}^{4 \times \frac{H}{8} \times \frac{W}{4}}$ is then split across the spatial dimension to extract the person part $z_0^p \in \mathbb{R}^{4 \times \frac{H}{8} \times \frac{W}{8}}$, we use the VAE decoder $D$ to transform the denoised latent representation $z_0^p$ back into the image space, producing the final output image $\widetilde{I}_p \in \mathbb{R}^{3 \times H \times W}$:

$$\widetilde{I}_p = D\left(Split\left(z_0, W\right)\right), \tag{5}$$

where $Split(\cdot, W)$ means split across the spatial dimension in width.

## 4 EXPERIMENTS

### 4.1 DATASETS

Our experiments are conducted on three public datasets: VITON-HD (Choi et al., 2021), Dress-Code (Morelli et al., 2022), and DeepFashion (Ge et al., 2019). VITON-HD comprises 13,679

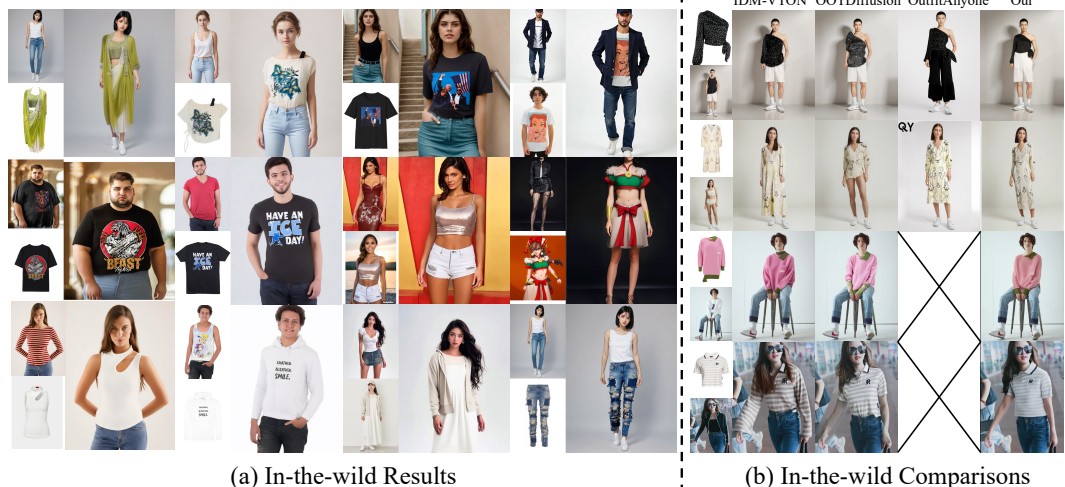

(a) In-the-wild Results | (b) In-the-wild Comparisons

Figure 6: Qualitative results and comparisons in in-the-wild scenarios. OutfitAnyone (Sun et al., 2024) only supports inference on its provided person images. Our method combines background, person, and garment more naturally in complex scenarios. Please zoom in for more details.

image pairs of upper, 11,647/2,032 training/testing pairs. DressCode is composed of 48,392/5,400 training/testing pairs with full-body person images and in-shop upper, lower, and dresses. Besides, we select 13,098/1,896 training/testing image pairs for the garment transfer task from the in-shop clothes retrieval benchmark of the DeepFashion dataset, which includes 52,712 high-resolution person images. For DressCode and DeepFashion datasets, we process clothing-agnostic masks using human parsing results from DensePose (Güler et al., 2018) and SCHP (Li et al., 2020) of LIP (Gong et al., 2017) and ATR (Liang et al., 2015) versions.

### 4.2 IMPLEMENTATION DETAILS

We utilize the inpainting and InstructPix2Pix (Brooks et al., 2023) version of StableDiffusion v1.5 (Rombach et al., 2021) as the base models for the mask-based and mask-free try-on models, respectively. We train two models for each version on the VITON-HD (Choi et al., 2021) and DressCode (Morelli et al., 2022) datasets separately for fair quantitative comparisons with previous methods. The AdamW (Loshchilov & Hutter, 2019) optimizer is employed with a batch size of 128 and a constant learning rate of $1e - 5$ for $16,000$ steps training under $512 \times 384$ resolution and DREAM $\lambda = 10$. Additionally, multi-task models are trained on the three datasets ($\sim 73$K image pairs) under $1024 \times 768$ resolution for 48,000 steps with an identical setup but a batch size of 32. All experiments are conducted on 8 NVIDIA A800 GPUs, which takes approximately 10 hours for 16K training steps.

### 4.3 METRICS

For paired try-on settings with ground truth in test datasets, we employ four widely used metrics to evaluate the similarity between synthesized images and authentic images: Structural Similarity Index (SSIM) (Wang et al., 2004), Learned Perceptual Image Patch Similarity (LPIPS) (Zhang et al., 2018), Frechet Inception Distance (FID) (Seitzer, 2020), and Kernel Inception Distance (KID) (Bińkowski et al., 2021). For unpaired settings, we use FID and KID to measure the distribution of the synthesized and real samples.

### 4.4 QUALITATIVE COMPARISON

Figure 5 (a) presents the try-on results of garments with complex patterns from the VITON-HD (Choi et al., 2021) dataset. While other methods often exhibit artifacts, loss of detail, and blurry text logos, CatVTON demonstrates its superiority by effectively handling texture positioning and occlusions and producing more photo-realistic results. Figure 5 (b) illustrates the comparison for different garment types (upper, lower, and dress) on full-body person images from the DressCode (Morelli et al., 2022) dataset. Our approach can generate results that are more consistent with the garment textures, length, and semi-transparent materials. We provided additional qualitative

Table 1: Quantitative comparison with other methods. We compare the metrics under paired and unpaired settings on the VITON-HD and DressCode datasets. The best and second-best results are demonstrated in **bold** and underlined, respectively.

| Methods | VITON-HD | | | | | | DressCode | | | | | |
|---|---|---|---|---|---|---|---|---|---|---|---|---|
| | Paired | | | | Unpaired | | Paired | | | | Unpaired | |
| | SSIM↑ | FID↓ | KID↓ | LPIPS↓ | FID↓ | KID↓ | SSIM↑ | FID↓ | KID↓ | LPIPS↓ | FID↓ | KID↓ |
| DCI-VTON (Gou et al., 2023) | 0.8620 | 9.408 | 4.547 | 0.0606 | 12.531 | 5.251 | - | - | - | - | - | - |
| StableVTON (Kim et al., 2023) | 0.8543 | 6.439 | 0.942 | 0.0905 | 11.054 | 3.914 | - | - | - | - | - | - |
| StableGarment (Wang et al., 2024c) | 0.8029 | 15.567 | 8.519 | 0.1042 | 17.115 | 8.851 | - | - | - | - | - | - |
| MV-VTON (Wang et al., 2024a) | 0.8083 | 15.442 | 7.501 | 0.1171 | 17.900 | 8.861 | - | - | - | - | - | - |
| GP-VTON (Xie et al., 2023) | 0.8701 | 8.726 | 3.944 | 0.0585 | 11.844 | 4.310 | 0.7711 | 9.927 | 4.610 | 0.1801 | 12.791 | 6.627 |
| LaDI-VTON (Morelli et al., 2023) | 0.8603 | 11.386 | 7.248 | 0.0733 | 14.648 | 8.754 | 0.7656 | 9.555 | 4.683 | 0.2366 | 10.676 | 5.787 |
| IDM-VTON (Choi et al., 2024) | 0.8499 | 5.762 | 0.732 | 0.0603 | 9.842 | 1.123 | 0.8797 | 6.821 | 2.924 | 0.0563 | 9.546 | 4.320 |
| OOTDiffusion (Xu et al., 2024) | 0.8187 | 9.305 | 4.086 | 0.0876 | 12.408 | 4.689 | 0.8854 | 4.610 | 0.955 | 0.0533 | 12.567 | 6.627 |
| CatVTON (Mask-Free) | 0.8701 | 5.888 | 0.513 | 0.0613 | 9.287 | 1.168 | **0.9016** | 4.779 | 1.297 | **0.0452** | 7.400 | 2.619 |
| CatVTON (Inpainting) | **0.8704** | **5.425** | **0.411** | **0.0565** | **9.015** | **1.091** | 0.8922 | **3.992** | **0.818** | 0.0455 | **6.137** | **1.403** |

Table 2: Comparison of GFLOPs, inference time, and memory usage across different methods.

| Methods | GFLOPs | | | | Inference Time(s) | | Memory Usage | |
|---|---|---|---|---|---|---|---|---|
| | $E_{text}$ | $E_{image}$ | ReferenceNet | UNet | 512×384 | 1024×768 | 512×384 | 1024×768 |
| OOTDiffusion (Xu et al., 2024) | 13.08 | 155.62 | 509.12 | 547.34 | 4.76 | 36.23 | 6854 M | 8892 M |
| IDM-VTON (Choi et al., 2024) | 110.04 | 155.62 | 1340.15 | 1163.98 | 12.96 | 17.32 | 17112 M | 18916 M |
| StableVTON (Kim et al., 2023) | - | 155.62 | 173.80 | 545.27 | 12.17 | 36.10 | 9828 M | 14176 M |
| CatVTON(Ours) | - | - | - | 973.59 | 2.58 | 9.25 | 3276 M | 5940 M |

results and comparisons in various in-the-wild scenes, as shown in Figure 6. Our method performs exceptionally well on fine patterns of garments, without altering or distorting the text and patterns on the garments. It can also accurately reproduce special clothing designs, producing realistic effects such as wrinkles, lighting, and shadows.

## 4.5 QUANTITATIVE COMPARISON

**Comparison of Effect.** We conducted the quantitative comparison of effect with several open-source try-on methods on the VITON-HD and DressCode datasets under both paired and unpaired settings as presented in Table 1. Our method outperformed all others across the metrics. GP-VTON (Xie et al., 2023), IDM-VTON (Choi et al., 2024), and OOTDiffusion (Xu et al., 2024) also showed good performance. GP-VTON, as a warping-based method, had advantages in SSIM and LPIPS but performed weaker in KID and FID. This result suggests that warping-based methods may focus more on ensuring structural and perceptual similarity but lack realism and detailed naturalness.

**Comparison of Efficiency.** Table 3 and Figure 2 (b) demonstrate the quantitative comparison of efficiency, including parameters, memory usage, and extra conditions for inference. Our method contains only two modules, VAE and UNet, with 899.06M parameters. Moreover, our trainable parameters are reduced by 10+ times compared to other methods. During inference, our method has a significant advantage in memory usage and does not require extra conditions such as pose or text, alleviating the burden of inference.

Table 2 presents the inference efficiency comparison of GFLOPs, inference speed, and memory usage across different methods at 512×384 and 1024×768 resolutions on a single NVIDIA A100 GPU. Inference was performed with a batch size of 1. For inference time, we averaged the results of 10 runs with the same input to ensure accuracy. GFLOPs were calculated using the *calflops* (xiaoju ye, 2023) library. These comparison results demonstrate that our model can be deployed on resource-constrained devices, such as consumer-level GPUs with less than 8 GB of VRAM, while maintaining significantly better inference speed compared to other models. However, deploying high-resolution image generation models on terminal devices, such as smartphones, remains an area that requires further exploration.

## 4.6 ABLATION STUDIES

**Trainable Module.** We evaluated three modules for training: (1) UNet, (2) transformer blocks, and (3) self-attention. As shown in Table 4, more training weights do not bring significant improvements in performance but increase the memory requirement and decrease the training speed. Slight advantages brought by additional training weights may be due to the increased trainable components, which allow the model to fit the data distribution more quickly. Besides, we trained a self-attention

Table 3: Detailed comparison of model efficiency. $UNet_{ref}$, $E_{text}$, and $E_{image}$ represent the ReferenceNet, text encoder, and image encoder, respectively. Compared to other diffusion-based methods, CatVTON uses fewer modules, reducing total parameters by about $2\times$ and trainable parameters by $10\times+$. CatVTON requires significantly less memory during inference and does not need additional conditions such as pose or text.

| Methods | Params (M) | | | | | | | Memory | Conditions | |
| | VAE | UNet | $UNet_{ref}$ | $E_{text}$ | $E_{image}$ | Total | Trainable | Usage(G) | Pose | Text |
|---|---|---|---|---|---|---|---|---|---|---|
| OOTDiffusion (Xu et al., 2024) | 83.61 | 859.53 | 859.52 | 85.06 | 303.70 | 2191.42 | 1719.05 | 10.20 | - | ✓ |
| IDM-VTON (Choi et al., 2024) | 83.61 | 2567.39 | 2567.39 | 716.38 | 303.70 | 6238.47 | 2871.09 | 26.04 | ✓ | ✓ |
| StableVTON (Kim et al., 2023) | 83.61 | 859.41 | 361.25 | - | 303.70 | 1607.97 | 500.73 | 7.87 | ✓ | - |
| StableGarment (Wang et al., 2024c) | 83.61 | 859.53 | 1220.77 | 85.06 | - | 2248.97 | 1253.49 | 11.60 | ✓ | ✓ |
| MV-VTON (Wang et al., 2024a) | 83.61 | 859.53 | 361.25 | - | 316.32 | 1620.71 | 884.66 | 7.92 | ✓ | - |
| CatVTON (Ours) | 83.61 | **815.45** | - | - | - | **899.06** | **49.57** | **4.00** | - | - |

Table 4: Ablation results of different trainable modules. More trainable modules slightly impact performance but increase memory usage and slow training. Extra text conditions harm performance. IPS (items per second) indicates training speed.

| Trainable Module | Paired | | | | Unpaired | | Trainable | Training | Training |
| | SSIM ↑ | FID ↓ | KID ↓ | LPIPS ↓ | FID ↓ | KID ↓ | Params (M) | IPS ↑ | Memory (M) |
|---|---|---|---|---|---|---|---|---|---|
| UNet | 0.8692 | **5.2496** | **0.4017** | **0.0550** | **8.8131** | **0.9559** | 815.45 | 3.21 | 14289 |
| Transformers | 0.8558 | 5.4496 | 0.4434 | 0.0558 | 8.8423 | 1.0082 | 267.24 | 4.10 | 9981 |
| Self Attention + Text | 0.8517 | 6.5744 | 1.0690 | 0.0772 | 9.6998 | 1.6683 | 49.57 | 4.50 | 8805 |
| Self Attention | **0.8704** | 5.4252 | 0.4112 | 0.0565 | 9.0151 | 1.0914 | **49.57** | **4.75** | **8451** |

version with text conditions in the same setting, and the results show a decrease in performance, indicating that text conditions are redundant in image-based try-ons. Training only the self-attention modules and removing unnecessary text conditions achieves a balance between model performance and efficiency. The IPS and memory statistics are calculated in a setting with a batch size of 1 to avoid the impact of other environmental factors.

**Classifier-Free Guidance.** To evaluate the effect of classifier-free guidance (CFG), we run inferences with CFG strengths of 0.0, 1.5, 2.5, 3.5, 5.0, and 7.5 while keeping all other parameters constant. Figure 7 (b) shows that increasing CFG strength enhances image detail and fidelity. However, beyond a strength of 3.5, the results developed severe color distortions and high-frequency noise, degrading visual quality. We found that a CFG strength between 2.5 and 3.5 produces the most realistic and natural results. A CFG strength of 2.5 is used for all the other experiments.

**DREAM.** $\lambda$ is a hyperparameter that controls the strength of DREAM. Specifically, $\lambda = \infty$ means DREAM is disabled. As shown in Figure 7 (a), a small $\lambda$ causes overly smooth images, while a large $\lambda$ introduces excessive high-frequency noise, reducing naturalness. Table 5 shows resutls trained with different $\lambda$ values on the VITON-HD dataset. SSIM increases with $\lambda$, while FID and LPIPS first improve and then degrade, highlighting a trade-off between reduced distortion and perceptual quality. We find that $\lambda = 10$ best balances naturalness with detail fidelity in our training.

Table 5: Ablation results of different $\lambda$ in DREAM on VITON-HD dataset. $\lambda=\infty$ means no DREAM. Increasing $\lambda$ improves perceptual quality (lower LPIPS, KID, and FID) but increases distortion (lower SSIM) in an empirical range.

| $\lambda$ | Paired | | | | Unpaired | |
| | SSIM ↑ | FID ↓ | KID ↓ | LPIPS ↓ | FID ↓ | KID ↓ |
|---|---|---|---|---|---|---|
| 0 | **0.8740** | 10.4534 | 3.8866 | 0.0692 | 14.1045 | 5.2824 |
| 1 | 0.8716 | 8.0983 | 2.1977 | 0.0646 | 11.7652 | 3.2942 |
| 10 | 0.8704 | 5.4252 | 0.4112 | **0.0565** | 9.0151 | 1.0914 |
| 20 | 0.8633 | 5.5861 | 0.4005 | 0.0620 | 9.0877 | 1.0416 |
| $\infty$ | 0.8614 | 5.5561 | **0.3657** | 0.0631 | **8.9114** | **1.0049** |

## 5 CONCLUSION

In this work, we present CatVTON, a virtual try-on diffusion model with lightweight architecture, efficient training, and streamlined inference. CatVTON achieves a compact structure with 899.06M parameters by removing unnecessary text-related modules, reducing model complexity significantly. CatVTON proposes a parameter-efficient training strategy to focus on the most essential components, specifically the self-attention layers, preserving high-quality virtual try-on performance while minimizing training costs with only 49.57M trainable parameters. During inference,

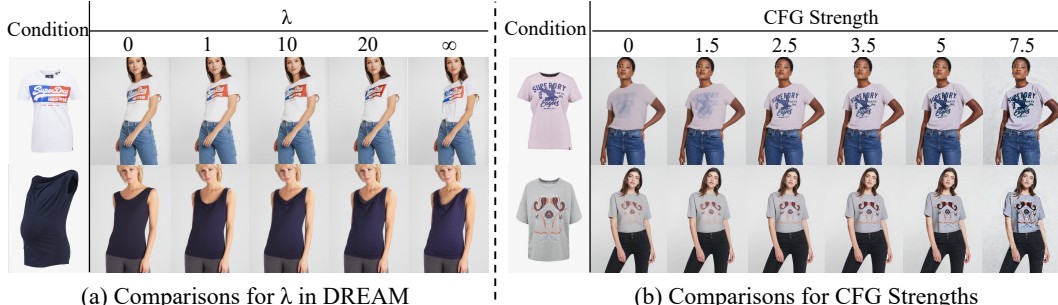

| (a) Comparisons for λ in DREAM | (b) Comparisons for CFG Strengths |

Figure 7: Visual comparisons for different $\lambda$ in DREAM and different CFG strengths. When $\lambda$ is too small, results are overly smooth and lack detail; when $\lambda$ is too large, results have excessive high-frequency details and appear unnatural. As the CFG strength increases, the details in the generated images increase, but beyond 3.5, it leads to severe color distortion and high-frequency noise.

CatVTON eliminates the need for additional information such as pose estimation, human parsing, and text-based inputs, reducing memory requirements and enhancing inference speed. Extensive experiments demonstrate that CatVTON delivers superior qualitative and quantitative results, outperforming state-of-the-art methods while maintaining a compact and efficient architecture. These findings underscore CatVTON's potential for practical applications and open new research directions in virtual try-on technology.

ACKNOWLEDGMENTS

Supported by Shenzhen Science and Technology Program No.GJHZ20220913142600001, National Natural Science Foundation Youth Basic Research Program (Doctoral Students) No.623B2100 and Nansha Key R&D Program under Grant No.2022ZD014.

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

# A  APPENDIX

## A.1  PRELIMINARY

**Latent Diffusion Models.** The core idea of Latent Diffusion Models (LDMs) (Rombach et al., 2021) is to map image inputs into a lower-dimensional latent space defined by a pre-trained Variational Autoencoder (VAE) (Kingma & Welling, 2022). In this way, Diffusion Models can be trained and inferred at a reduced computational cost while retaining the capability to generate high-quality images. The components of LDMs are primarily a denoising UNet $E_\theta(\circ, t)$ and a VAE which consists of an encoder $\mathcal{E}$ and a decoder $\mathcal{D}$. Given an input $x$, the training of LDM is carried out by minimizing the following loss function:

$$L_{LDM} := \mathbb{E}_{\varepsilon(x),\epsilon\sim\mathcal{N}(0,1),t} \left[ \|\epsilon - \epsilon_\theta(z_t,t)\|_2^2 \right], \tag{6}$$

where $t \in \{1,...,T\}$ denotes the timestep of the forward diffusion process. In the training phase, the latent representation $z_t$ is readily derived from $\mathcal{E}$ with the added Gaussian noise $\epsilon \sim \mathcal{N}(0,1)$. Subsequently, the latent samples, drawn from the distribution $p(z)$, are translated back into the image domain with just one traversal of $\mathcal{D}$.

**Diffusion Rectification and Estimation-Adaptive Models (DREAM).** DREAM (Zhou et al., 2024) is a training strategy designed to skillfully navigate the trade-off between minimizing distortion and preserving high image quality in image super-resolution tasks. Specifically, during training, the diffusion model is used to predict the added noise as $\epsilon_\theta$. This $\epsilon_\theta$ is then combined with the original added noise $\epsilon$ to obtain $\hat{\epsilon}$, which is used to compute $\hat{z}_t$:

$$\hat{z}_t = \sqrt{\overline{\alpha}_t} z_0 + \sqrt{1 - \overline{\alpha}_t}(\epsilon + \lambda\epsilon_\theta), \tag{7}$$

where $\lambda$ is a parameter to adjust the strength of $\epsilon_\theta$ and $\overline{\alpha}_t = \prod_{i=1}^{t} 1 - \beta_i$ with the variance scheduler $\{\beta_t \in (0,1)\}_{t=1}^{T}$. The training objective for DREAM can be expressed as:

$$L_{DREAM} := \mathbb{E}\varepsilon(x), \epsilon, \epsilon\theta \sim \mathcal{N}(0,1), t \left[ |(\epsilon + \lambda\epsilon_\theta) - \epsilon_\theta(\hat{z}_t, t)|_2^2 \right]. \tag{8}$$

DREAM enhances training efficiency and accuracy, although it requires an additional forward pass before the training prediction process, slightly slowing down the training process.

## A.2  IMPLEMENTATION DETAILS

### A.2.1  MASK-FREE DATASET

The construction of the mask-free dataset is based on the VITON-HD (Choi et al., 2021), DressCode (Morelli et al., 2022), and DeepFashion (Ge et al., 2019) datasets. We use the mask-based model to randomly generate pseudo-data for constructing mask-free paired data in the same garment category (uppers, lowers, and dresses). To ensure the accuracy of the masks, we employ multiple human parsing models (ATR(Liang et al., 2015), LIP(Gong et al., 2017)) and body part information from DensePose (Güler et al., 2018) to cross-validate the required mask regions. Additionally, convex hull and pooling operations are applied to ensure no information leakage in areas outside the mask. This comprehensive approach guarantees the quality of the generated data required for training the mask-free model, thereby enabling it to focus on the try-on regions.

### A.2.2  HARDWARE ENVIRONMENT

We conducted our experiments on a Linux server with an x86 architecture. It is equipped with an Intel Xeon CPU and 8 NVIDIA A800 GPUs, each with 80GB of VRAM.

### A.2.3  SOFTWARE ENVIRONMENT

Our work is implemented based on the PyTorch deep learning framework, with the version being 2.1.2. The code for the diffusion model is modified and implemented based on HuggingFace's Diffusers library.

### A.2.4  CONCATENATION ALONG X/Y-AXIS

During training, we experimented with the direction of spatial concatenation (along the x-axis or y-axis). Theoretically, for convolutional neural networks and Transformers without positional embeddings, the direction of spatial concatenation—whether along the x-axis or y-axis—should make no difference. Our experimental results are consistent with this theory; training with either x-axis or y-axis concatenation can produce normal results. Moreover, a model trained with x-axis concatenation can also yield normal results when using y-axis concatenation during inference.

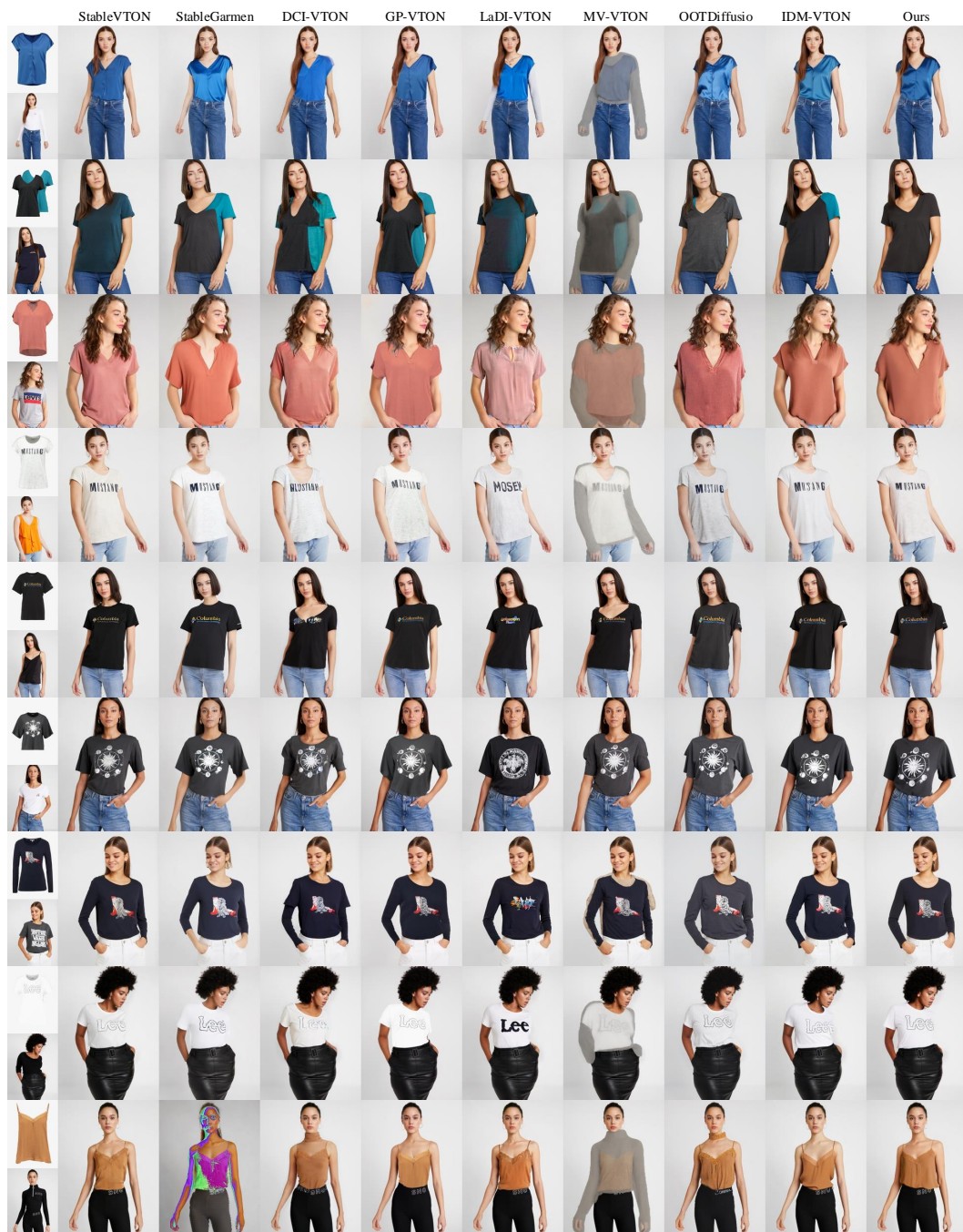

Figure 8: More visual comparisons on the VITON-HD dataset with baseline methods. Please zoom in for more details.

## A.3 MORE VISUAL COMPARISONS

### A.3.1 VTION-HD DATASET

Figure 8 presents additional virtual try-on results on the VITON-HD (Choi et al., 2021) test dataset, where our method demonstrates an advantage in preserving the details of text, patterns, and logos, and can adaptively fuse with the target person while maintaining a reasonable scale. In addition, CatVTON exhibits a more natural representation of garment designs such as sleeves and collars.

### A.3.2 DRESSCODE DATASET

| Methods | Paired | | | | Unpaired | |
|---|---|---|---|---|---|---|
| | SSIM ↑ | FID ↓ | KID ↓ | LPIPS ↓ | FID ↓ | KID ↓ |
| StableGarment Wang et al. (2024c) | 0.8029 | 15.567 | 8.519 | 0.1042 | 17.115 | 8.851 |
| MV-VTON Wang et al. (2024a) | 0.8083 | 15.442 | 7.501 | 0.1171 | 17.900 | 8.861 |
| LaDI-VTON Morelli et al. (2023) | 0.8603 | 11.386 | 7.248 | 0.0733 | 14.648 | 8.754 |
| DCI-VTON Gou et al. (2023) | 0.8620 | 9.408 | 4.547 | 0.0606 | 12.531 | 5.251 |
| OOTDiffusion Xu et al. (2024) | 0.8187 | 9.305 | 4.086 | 0.0876 | 12.408 | 4.689 |
| GP-VTON Xie et al. (2023) | 0.8701 | 8.726 | 3.944 | 0.0585 | 11.844 | 4.310 |
| StableVITON Kim et al. (2023) | 0.8543 | 6.439 | 0.942 | 0.0905 | 11.054 | 3.914 |
| CatVTON (DiT) | **0.9118** | **5.786** | **0.939** | **0.0393** | **10.019** | **1.864** |

Table 6: Quantitative comparison of DiT version with other methods on VITON-HD Choi et al. (2021) dataset. The best and second-best results are demonstrated in **bold** and underlined, respectively.

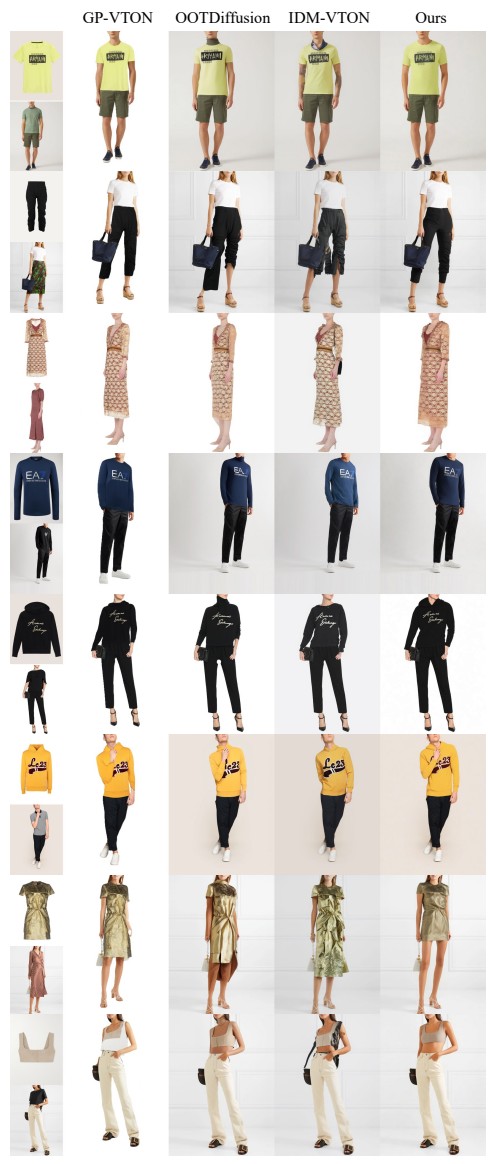

Figure 9: More visual comparisons on the Dress-Code dataset with other baselines. Please zoom in for more details.

The visual comparisons on the DressCode (Morelli et al., 2022) test dataset are further displayed in Figure 9, where our method can better recognize and match the lengths of different types of clothing and can generate more coherent patterns for situations such as arm occlusion.

### A.4 Transferability

To extend the transferability of our proposed method, we conducted experiments using HunyuanDiT (Li et al., 2024) as the pre-trained model on the VITON-HD (Choi et al., 2021) dataset. Table 6 presents the comparative results of our approach within the HunyuanDiT framework on the VITON-HD dataset. Although DiT converges more slowly compared to UNet and has not fully fitted the data, the results of our DiT-based version still outperform most existing methods.

### A.5 Limitations & Social Impacts

While leveraging LDM (Rombach et al., 2021) as the backbone for generation, our model faces certain limitations. Images decoded by VAE may exhibit detail loss and color discrepancies, particularly at a 512×384 resolution. Additionally, the effectiveness of the try-on process is contingent upon the accuracy of the provided mask; an inaccurate mask can significantly degrade the results. Based on Stable Diffusion v1.5, our pre-trained model was trained on large-scale datasets that include not-safe-for-work (NSFW) content. Consequently, retaining most of the original weights means our model may inherit biases from the pre-trained model, potentially generating overly explicit images of people.

