# OpenReview forum: "CatVTON: Concatenation Is All You Need for Virtual Try-On with Diffusion Models"
_ICLR.cc/2025/Conference — ICLR 2025 Poster_

### Official Review · Reviewer_iWwH · 2024-10-27

**Soundness:** 2
**Presentation:** 2
**Contribution:** 3
**Rating:** 5
**Confidence:** 4

**Summary:**

**Post-discussion period**

---

I will maintain my current score of 5, as the issues I raised in my review remain unresolved:

+ Discrepancies in Quantitative Results: There are significant discrepancies between the reported results in the submission and the original baseline papers. This inconsistency makes the comparison less convincing. To address this, I suggested that the authors use the original evaluation scripts from the baseline papers alongside their own settings, but it is disregarded. This would provide a fairer comparison, particularly given the considerable variance observed in metrics like LPIPS, FID, and KID.

+ Architectural and Textual Inconsistencies: There is a minor but important inconsistency between the overall architecture figure and the description in the main text. Specifically, the process of input concatenation, whether in pixel-space or latent-space, is unclear. This discrepancy raises questions about whether the results are indeed identical to what was claimed.

For me, if there are unclear aspects in a submission and the authors disregard suggestions such as using original scripts for reevaluation, I cannot raise the score or recommend acceptance unless these concerns are fully addressed. The final decision I would let the effort of ACs.

---

This work introduces CatVTON, a diffusion-based virtual try-on model that enhances efficiency by removing complex modules and preprocessing steps typical in other approaches. Diffusion-based virtual try-on methods usually require additional encoding, extensive training, and preprocessing, which CatVTON addresses by simplifying the architecture to a VAE and denoising UNet, reducing parameters to 899.06M. Through analysis, the authors find that self-attention modules are essential for adapting diffusion models to this task, enabling high-quality outputs with only 49.57M training parameters. CatVTON further reduces inference complexity, eliminating pre-processings like pose estimation and human parsing, which cuts memory usage by over 49%. Extensive testing demonstrates CatVTON’s strong performance and generalization, even with a limited 73K sample dataset, achieving better results than baseline methods in both controlled and real-world scenarios. Experiments are conducted on two datasets: VITON-HD and DressCode.

**Strengths:**

+ Removing the redundant reference Unet of most existing work is a good direction as it makes the model lightweight and simplified.
+ Results on two benchmark datasets VTON-HD and DressCode show good performance.

**Weaknesses:**

**Concerns**
+ Inference speed comparison is missing since the proposed model is more lightweight, it is more beneficial to see this metric.

+ Many works (AnyDoor, MimicBrush, Uni-ControlNet) show that a strong pre-trained image encoder (CLIP, DINOv2, Unet) would provide more priors and good features for better imager generation with better details, but this work shows that removing all of them, and using only VAE outputs can give even better performance. Which in my opinion, it might not be very reasonable. A valid explanation is needed to this point and also I believe that whether the code of this paper is publicly available for the community to verify its validity and reproducibility is crucial.

+ The detail of all numeric numbers in Table 1, and Table 2 should be clarified. Were all of them reproduced by the author or were they quoted from somewhere else? Looking at the original papers (OOTDiffusion and others) I find that they report much higher numbers compared to the numbers shown in this submission. It raised questions about the correctness and what made the differences since they used also the same dataset.

+ "DREAM \lambda", just putting this terminology here is not very informative, it is quite hard to understand what is that parameter \lambda and what is DREAM here without some more detailed explanations.

For the whole paper, the current form seems to use all \cite{} making it very messy for all references. I think the use of \citep{} in latex would produce a more correct presentation of citations for many parts of the paper. Using \cite{} for cases where the citing author is subject (S), but \citep{} for other cases when referring to the paper.

[1] AnyDoor: Zero-shot Object-level Image Customization, CVPR 2024

[2] MimicBrush: Zero-shot Image Editing with Reference Imitation, arXiv:2406.07547

[3] Uni-ControlNet: All-in-One Control to Text-to-Image Diffusion Models, EnruIPS 2023

**Questions:**

Q1. What is the IPS metric? It is better to provide its definition and description of it if it is newly proposed or referred to if needed.

---

> ### Author Response · Authors · 2024-11-25
>
> We hope our response has addressed your concerns. If there are any other issues you'd like to discuss, please don't hesitate to let us know.

---

### Official Review · Reviewer_F9hE · 2024-10-28

**Soundness:** 4
**Presentation:** 4
**Contribution:** 4
**Rating:** 8
**Confidence:** 5

**Summary:**

This paper aims to efficiently adapt pre-trained diffusion models to high-quality virtual try-on. It introduces efficiency improvements in three key areas: model architecture, parameter-efficient training, and simplified inference, and analyzes the motivation and feasibility from the perspective of image representation interaction, supported by experimental validation. Extensive qualitative and quantitative experiments are conducted on VITON-HD and DressCode datasets, along with ablation studies on key components of proposed method. Compared to other virtual try-on methods, this method demonstrates significant advantages in model parameters, training efficiency, and inference costs while achieving better performance. Despite being trained on small-scale public datasets, results and comparisons in in-the-wild scenarios demonstrate the method's generalization capabilities.

**Strengths:**

1. Originality: This paper focuses on efficiency optimizations in diffusion models for virtual try-on. By simplifying the model architecture with a concatenation operation and identifying key modules through an analysis of image representation interactions, it achieves parameter-efficient training. Additionally, by streamlining preprocessing in inference, it achieves efficient inference without extra conditions. The approach and model architecture differ significantly from other virtual try-on models, addressing the efficiency issues of previous work. The paper offers new insights into the application of image representation interactions in generative models, demonstrating innovation.

2. Quality: This paper conducts extensive comparative and ablation experiments to validate its performance. Qualitative and quantitative analyses are provided on two major open-source datasets, DressCode and VITON-HD, in comparison with recent virtual try-on methods. Quantitatively, the proposed model shows significant advantages in model size, trainable parameters, and inference memory usage, while surpassing other methods in FID, KID, SSIM, and LPIPS metrics. Visually, the model generates highly realistic details such as text logos and patterns, and performs well in in-the-wild virtual try-on scenarios.

3. Clarity: The writing is well-structured, with a clear and logical flow of content. Plenty of figures, tables, and formulas are used to explain and analyze key insights and experimental details. The writing is smooth, with no obvious grammatical errors or inconsistencies.

4. Significance: This work offers insights and contributions to the application of diffusion models in virtual try-on tasks. Its lightweight architecture and parameter-efficient training significantly reduce model training costs and may be transferable to other downstream tasks. The model's low memory usage during inference and simplified inference process reduce hardware requirements, making it more cost-effective to deploy.

**Weaknesses:**

1. The description of the proposed mask-free training method is not specific and detailed enough, more information should be provided such as the dataset details used for this part of the training and the exact process of pseudo-data construction.

2. The Concatenation operation applied to garment and person features lacks a more explicit and detailed discussion, such as its advantages over encoder features like CLIP or DINO.

3. The transferability of the proposed efficient virtual try-on method based on Stable Diffusion is not discussed. For example, migrating this training framework to more advanced image generation diffusion models, such as DiT, to verify the generality of this framework.

4. Some inappropriate formatting issues, such as inconsistent font sizes in tables. The font in Table 1 and Table 4 is relatively small, and there is excessive whitespace in Table 4. The images in Figure 5 and Figure 6, which are used to display generated results, are arranged too tightly.

**Questions:**

1. Can the proposed efficient training method be transferred to other more novel diffusion models, such as DiT?

2. Will the generated dataset used for mask-free training be open-sourced to support future research?

3. Provide more details about Mask-free training, such as dataset construction and how it differs from Mask-based model training, visual comparison of its results with Mask-based model results, etc.

4. Adjust the layout of the tables and figures to make them more uniform and standardized.

---

> ### Comment · Reviewer_F9hE · 2024-11-25
>
> Thanks for the author's reply. The author's reply answered my doubts.

---

> > ### Author Response · Authors · 2024-11-25
> >
> > Thank you for your feedback. We're glad our response addressed your concerns.

---

### Official Review · Reviewer_oVwb · 2024-11-03

**Soundness:** 3
**Presentation:** 3
**Contribution:** 3
**Rating:** 6
**Confidence:** 4

**Summary:**

CatVTON is a virtual try-on model that achieves high-quality and efficient garment transfer without relying on additional encoders or complex preprocessing. Unlike traditional virtual try-on approaches, CatVTON simplifies the architecture by using only a VAE and a single, lightweight UNet with a self-attention module, reducing the parameter count to 899.06M. Moreover, this model directly concatenates the person and garment images along spatial dimensions, requiring only 49.57M trainable parameters and cutting down inference memory usage by over 49% compared to other methods. Extensive experiments demonstrate that CatVTON achieves superior results in both controlled and in-the-wild settings.

**Strengths:**

- CatVTON proposes a lightweight architecture, which highly reduces computational cost during the model training. Meanwhile, it only requires 49.57M training parameters.

- The model generalises well in diverse real-world scenarios, performing effectively even with limited training data.

- CatVTON is able to achieve realistic try-on results with accurate garment alignment and detail preservation.

**Weaknesses:**

- While the authors present CatVTON as a lightweight virtual try-on diffusion model and it only requires very few training parameters, it would be beneficial to include results on inference speed and computational cost. Additionally, demonstrating the model’s deployability on embedded devices could enhance understanding of its practical applications.

- The mask-free model is derived from a trained mask-based model. It would be better to discuss potential error accumulation in this transition.

- CatVTON introduces a model design and training strategy aimed at creating a lightweight model with fewer training parameters. It would be useful to consider whether this approach could be extended to other virtual try-on pipelines, particularly non-diffusion-based methods.

**Questions:**

- Can you provide additional quantitative results on CatVTON’s inference speed and computational cost, and is the model deployable on embedded or resource-constrained devices?

- Given that the mask-free model builds on a trained mask-based model, have you observed any error accumulation in this process, and if so, what techniques might mitigate such errors?

- Could the lightweight model design and parameter-efficient training strategy you propose for CatVTON be applicable to other virtual try-on frameworks, including non-diffusion-based methods?

---

> ### Author Response · Authors · 2024-11-25
>
> We hope our response has addressed your concerns. If there are any other issues you'd like to discuss, please don't hesitate to let us know.

---

### Official Review · Reviewer_mw7R · 2024-11-04

**Soundness:** 3
**Presentation:** 3
**Contribution:** 2
**Rating:** 6
**Confidence:** 5

**Summary:**

The study re-evaluates the necessity of additional modules in virtual try-on methods based on diffusion models and proposes CatVTON, a simple and efficient model. CatVTON transfers garments of arbitrary categories to target individuals by concatenating them as inputs, removing redundant encoding modules and steps. It features a lightweight network with just 899.06M parameters, efficient training with only 49.57M parameters through self-attention modules, and simplified inference that requires only person image and garment reference, reducing memory usage by 49%+. Experiments show that CatVTON achieves superior results and generalizes well in real-world scenarios, despite being trained on a limited public dataset.

**Strengths:**

+ 1. This paper proposes CatVTON, a lightweight virtual try-on diffusion model with only 899.06M parameters, that achieves high-quality results by simply concatenating garment and person images as inputs, eliminating the need for extra image encoders, ReferenceNet, and text-conditioned modules.

+ 2. This paper introduces a parameter-efficient training strategy to transfer pre-trained diffusion models to virtual try-on tasks while preserving prior knowledge by training necessary modules with only 49.57M parameters

+ 3.  This paper simplifies the inference process by eliminating the need for extra pre-processing of input images and leveraging the robust priors from pre-trained diffusion models to infer all necessary information, reducing memory usage by 49%+ compared to other diffusion-based baselines

**Weaknesses:**

- 1. The effectiveness of the try-on process relies on the accuracy of the provided masks; inaccurate masks may significantly degrade the quality of the results. When the authors conducted mask-free experiments, how did they ensure that the model paid attention to the try-on areas?

- 2. Some of the data in Table 1 seem to differ from those in the original paper, why?

- 3. Based on Table 3, it seems that choosing to train either the Unet or Transformers module can achieve better results, albeit with higher memory demands and slower training. How to balance this relationship?

- 4. Please provide the training code to demonstrate reproducibility.

- 5. The method is simple and effective, however, the entire process is overly engineered, lacking rigorous theoretical validation and support in a conference like ICLR.

**Questions:**

Please refer to "Weaknesses."

---

> ### Author Response · Authors · 2024-11-25
>
> We hope our response has addressed your concerns. If there are any other issues you'd like to discuss, please don't hesitate to let us know.

---

> > ### Comment · Reviewer_mw7R · 2024-11-25
> >
> > The author has addressed my main concerns, so I don't have any further questions.

---

### Meta-Review · Area_Chair_pojz · 2024-12-15

**Metareview:**

In this paper, the authors propose CatVTON, a simple and efficient virtual try-on diffusion model using a VAE and denoising UNet with spatial concatenation, reducing memory usage and training parameters. The paper demonstrated competitive performance on standard benchmarks and in real-world scenarios, with strong generalization and efficient inference. However, the reviewers raised concerns about reproducibility due to discrepancies in evaluation metrics compared to prior works, insufficient explanation of the concatenation's advantages, and a lack of detailed theoretical validation.

After the author-reviewer discussion period, most of the reviewers confirmed their concerns had been addressed. However, Reviewer iWwH still has two important remaining concerns:
- Discrepancies in quantitative results: there are significant discrepancies between the reported results in the submission and the original baseline papers.
- Architectural and textual inconsistencies: there is a minor but important inconsistency between the overall architecture figure and the description in the main text.

Considering most of the reviewers are positive towards this submission, and Reviewer iWwH is not strongly against accepting it, I recommend acceptance. However, the discussions between the reviewers and authors, especially the concerns from Reviewer iWwH, should be included in the final version.

**Additional Comments On Reviewer Discussion:**

During the author-reviewer discussion period, most of the reviewers confirmed their concerns had been addressed. However, Reviewer iWwH still has two important remaining concerns:
- Discrepancies in quantitative results: there are significant discrepancies between the reported results in the submission and the original baseline papers.
- Architectural and textual inconsistencies: there is a minor but important inconsistency between the overall architecture figure and the description in the main text.

Considering most of the reviewers are positive towards this submission, and Reviewer iWwH is not strongly against accepting it, I recommend acceptance. However, the discussions between the reviewers and authors, especially the concerns from Reviewer iWwH, should be included in the final version.

---

### Decision · Program_Chairs · 2025-01-22

Accept (Poster)